# Determining the threshold of issuing flash flood warnings based on people's response process simulation

Ruikang Zhang [a, b], Dedi Liu [a, b, c*], Lihua Xiong [a, b], Jie Chen [a, b], Hua Chen [a, b], Jiabo Yin [a, b]

[a] State Key Laboratory of Water Resources Engineering and Management, Wuhan University, Wuhan, China

[b] Hubei Provincial Key Lab of Water System Science for Sponge City Construction, Wuhan University, Wuhan, China

[c] Department of Earth Science, University of the Western Cape, Robert Sobukwe Road, Bellville 7535, Republic of South Africa

* Correspondence to Dedi Liu: dediliu@whu.edu.cn

**Abstract:** The effectiveness of flash flood warnings depends on the people's response processes to the warnings. And false warnings and missed events cause the people's negative responses. It is crucial to find a way to determine the threshold of issuing the warnings that reduces the false warning ratio and the missed event ratio, especially for uncertain flash flood forecasting. However, most studies determine the warning threshold based on the natural processes of flash floods rather than the social processes of warning responses. Therefore, an agent-based model (ABM) was proposed to simulate the people's response processes to the warnings. And a simulation chain of "rainstorm probability forecasting - decision on issuing warnings - warning response processes" was conducted to determine the warning threshold based on the ABM. Liulin Town in China was selected as a case study to demonstrate the proposed method. The results show that the optimal warning threshold decreases as the forecasting accuracy increases. And as the forecasting variance or the variance of the forecasting variance increases, the optimal warning threshold decreases (increases) for low (high) forecasting accuracy. Adjusting the warning threshold according to the people's tolerance levels of the failed warnings can improve warning effectiveness, but the prerequisite is to increase the forecasting accuracy and decrease the forecasting variance. The proposed method provides valuable insights into the determination of warning threshold for improving the effectiveness of flash flood warnings.

**Keywords:** Threshold of issuing warnings; Flash flood warnings; People's response processes; Evacuation; Agent-based model

## 1. Introduction

With the intensification of climate change and human activities (Slater et al., 2021), flash floods have become one of the most serious disasters threatening economic and social security (Borga et al., 2019). Flash flood warning has been taken as an effective and economical means of preventing flash flood disasters (Yin et al., 2023). By issuing warnings before the occurrence of flash floods, people are advised to or ordered to evacuate for reducing the casualties. However, the people's responses to the warnings are complex processes including receiving the warnings, understanding the warnings, trusting the warnings, and personalizing the flood risk (Mileti, 1995; Parker et al., 2009). And these complex processes might hinder the evacuation and undermine the effectiveness of the warnings (Cools et al., 2016). To improve the effectiveness of flash flood warnings, extensive studies have been done to pursue higher accuracy and longer lead time of flash flood forecasting (Han and Coulibaly, 2017; Lei et al., 2018). Unfortunately, the people's responses to the warnings have rarely been explored and have become a bottleneck in improving the effectiveness of the warnings and reducing casualties (Bodoque et al., 2019; Wang et al., 2022).

The people's negative responses to the warnings have been mainly attributed to the uncertainties of the flash flood forecasting and the warnings. The uncertainties of flash flood forecasting are from the uncertainties of meteorological forecasting, observation data, initial conditions, hydrological and hydraulic model structure, model parameters, and so on (Boelee et al., 2019). To describe the uncertainties of flood forecasting, a probabilistic flood forecasting was proposed and had been widely applied in the issuing warnings by the disaster prevention administrators (Krzysztofowicz, 2001). If the probability of flash flood disasters from the probabilistic flood forecasting exceeds a preset threshold, the procedure of the issuing warning will be triggered (Coccia and Todini, 2011; Todini, 2017). If the threshold is set low, even a low forecasted probability of flash flood disasters can exceed the threshold, and lots of warnings with only the low probability of flash flood disaster will be issued, resulting in an increase in the false warning ratio. In contrast, if the threshold is set high, only the flash flood disasters with high forecasted probability can be warned, and some flash flood disasters with not low probability will be missed, leading to an increase in the missed event ratio (Potter et al., 2021). These two increases from both the false warning ratio and the missed event ratio can decrease the people's responses to the warnings and expand the

casualties. Simmons and Sutter (2009) conducted a statistical analysis of tornado data
from 1986 to 2004, and they found that tornadoes with a higher false warning ratio
killed and injured more people. LeClerc and Joslyn (2015) explored the cry wolf effect
in weather-related decision making through a controlled experimental approach. And
their experiments revealed that the decreasing false warning ratio could increase
people's trust in the warnings when the trust level was in the medium range, while both
too high and too low false warning ratios led to inferior decision making. Ripberger et
al. (2015) found that the false warning ratio and the missed event ratio significantly
reduced people's trust in the National Weather Service, and suppressed their positive
responses via a large regional survey. However, it is impossible to simultaneously
reduce the false warning ratio and the missed event ratio at a certain level of forecasting,
as there is a trade-off between these two ratios as described above. Therefore, it is
crucial to find a way to determine an appropriate threshold that balances the false
warning ratio and the missed event ratio for improving the positive warning responses
and reducing the disaster casualties.

Extensive methods have been proposed to determine the threshold of issuing flood

warnings for balancing the false warning ratio and the missed event ratio (Duc Anh et
al., 2020; Ke et al., 2020; Ramos Filho et al., 2021; Tekeli and Fouli, 2017; Young et
al., 2021). The methods have gradually evolved from fixed threshold determination
methods to dynamic threshold determination methods, and from data-driven methods
to simulation-based methods (Cheng, 2013). However, these methods only determined
the threshold of issuing warnings based on the natural processes of flash floods, while
ignoring the social processes of warning responses. The goal of flash flood warnings is
to stimulate the people's responses to the warnings for reducing casualties. Even a
reliable warning cannot be effective without people's positive responses to it. To our
best knowledge, there are very few methods to determine the threshold based on
people's response process simulation. Roulston and Smith (2004) generalized the
warning release into an improved classical binary cost-loss problem, where the people's
warning response level was expressed as a function of false warning ratio, and this
warning response level variable was included in the cost-loss analysis. And the
threshold of issuing warnings was derived with the goal of minimizing the cost loss
ratio under different scenarios. Sawada et al. (2022) proposed a stylized model that
coupled natural and social systems to determine the threshold of issuing warnings. In
this stylized model, the warning response level was attributed to be influenced by both

the success rate of the warning and the flood experience, and then was mapped to flood losses through an empirical equation. However, these studies only described the warning response level through empirical equations or conceptual models, instead of describing the warning response processes through process-based models. To reflect the characteristics of flash flood disaster prevention and the flash flood warning responses, it is necessary to simulate the people's response processes of receiving warnings, making evacuation decisions, implementing evacuation, and being submerged by flash floods (or reaching shelters).

Agent-based model (ABM) is a modeling framework for complex systems by simulating the dynamic interactions between automatic decision-making agents and between these agents and the environment in a distributed micro level (Janssen and Ostrom, 2006). As the warning responses are related to a learning process, and also to personal flood experience and risk perception, ABM is suitable for understanding the dynamic processes through simulating the individual decision-making (Anshuka et al., 2022). Additionally, ABM can describe the spatially explicit social-hydrological processes, such as the dissemination of warning information, the selection of evacuation routes, and the distribution of flash flood inundation (Sivapalan and Bloeschl, 2015). Thus, ABM is an effective tool for simulating the people's response processes to flash flood warnings (Du et al., 2017; Du et al., 2023; Yang et al., 2018; Zhuo and Han, 2020).

The objective of this study includes two parts. Firstly, to simulate people's response processes to flash flood warnings and reveal the impact of the warning information weight given by people on the effectiveness of warnings, this study aims to develop a process-based ABM that combines natural and social processes (section 2.1). Secondly, to determine the threshold of issuing warnings (called warning threshold hereafter) based on the social processes of warning responses, this study attempts to propose a simulation chain of "rainstorm probability forecasting - decision on issuing warnings - warning response processes" based on the ABM (section 2.2). Through the proposed simulation framework for determining the warning threshold, we will examine the uncertainties in flash flood forecasting that affect the determination of warning thresholds and the joint impact of forecasting skills and people's tolerance levels of failed warnings on the warning threshold determination. Liulin Town in China is selected as a case study to demonstrate the proposed method, and to provide valuable insights into the determination of warning threshold for improving the effectiveness of

flash flood warnings.

## 2.  Methodology

A modeling framework is proposed to determine the warning threshold based on
people's response processes. The modeling framework includes the development of an
ABM and its surrogate model for simulating the people's response processes to flash
flood warnings and a chain simulation of "forecasting – warning - response" (see
**Figure 1**). First, rainstorm probability forecasting is performed according to actual
rainfall. And then the warning administrators make decisions to issue warnings based
on the rainstorm probability forecasting and warning thresholds. If it is decided to issue
warnings, the warning information and the actual rainfall jointly drive the surrogate
model of ABM to simulate the people's response processes. Finally, the casualty rate is
estimated and the warning threshold that minimizes the casualty rate can be determined
based on the proposed modeling framework.

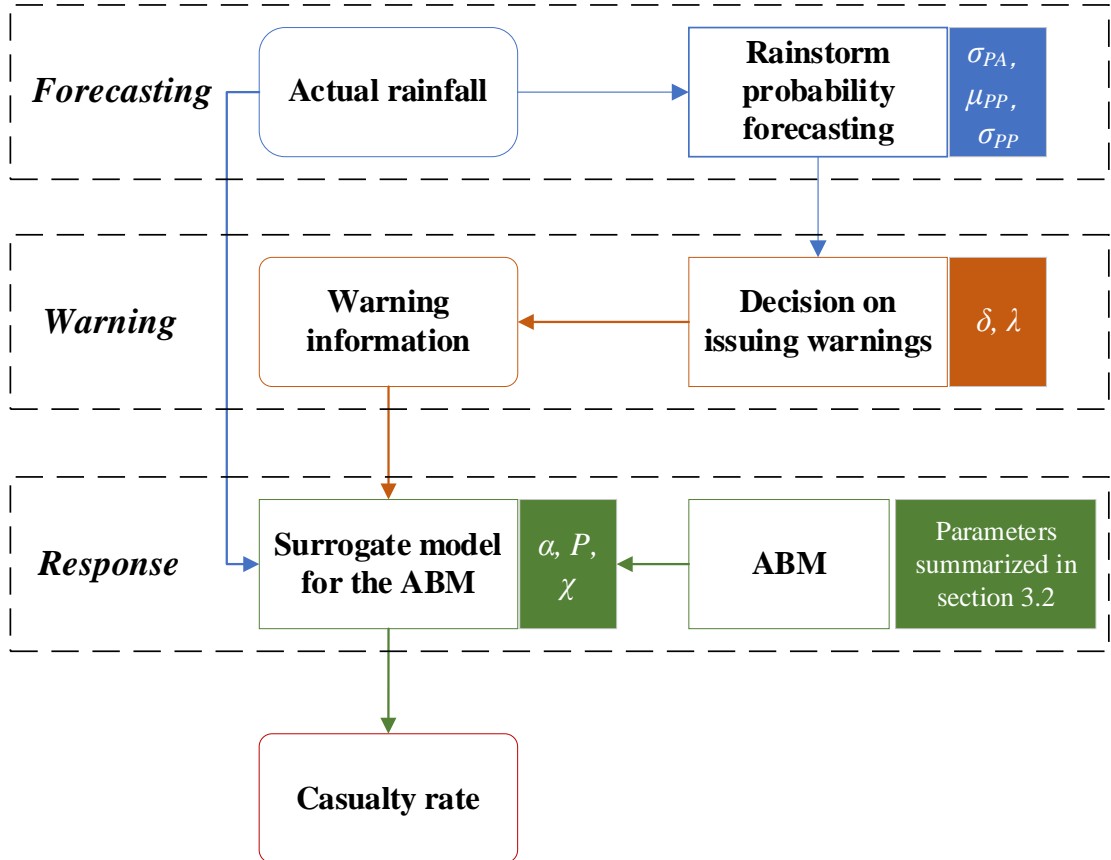


**Figure 1.** The proposed modeling framework for determining the warning threshold
based on people's response processes (the parameters in a simulation step are indicated
by a rectangular box with the corresponding color background)

## 2.1. An ABM development for simulating people's response processes to flash flood warnings

To simulate the people's response processes to flash flood warnings (i.e., including the receiving warnings, the making evacuation decisions, the implementing evacuation, and the being submerged by flash floods/the reaching shelters), an ABM is developed by coupling social and natural sub-systems.

### 2.1.1. Agents and their environments in the ABM

There are two types of agents in the ABM: resident and authority. The resident agents refer to the people threatened by flash floods. After receiving flash flood warnings, the agents will decide whether and when to evacuate. If they decide to evacuate, they will move along the roads towards the shelters. After issuing the warnings, the flash flood will occur and might wash away the agents who have not successfully arrived at shelters. The probability of casualties can be estimated based on the velocity and the depth of the flash flood. The authority agents represent the local authorities that mandate to prevent the flash flood disasters.

The environment in the ABM are the residences, road networks, shelters, and floodwater. The residence agents are initially randomly distributed in the residences. The resident agents who have decided to evacuate will move along the road network instead of freely moving within the ABM area. The shelters are the destinations for evacuation. The flash flood water not only affects the evacuation decisions and behaviors of the resident agents but also causes casualties to the resident agents.

### 2.1.2. Sub-modules of the ABM

*Early warning sub-module*. Early warning sub-module simulates the process of issuing warnings. Owing to the uncertainties of flash flood forecasting, there are multiple stages of warning in a warning system. Rainstorm red, ready-to-evacuate, and immediate-evacuation warnings are successively issued in the ABM. The times of issuing these three warnings are determined by three parameters: lead time of rainstorm red warning (indicated as $lead\text{-}time\text{-}w1$), ready-to-evacuate warning (indicated as $lead\text{-}time\text{-}w2$), and immediate-evacuation warning (indicated as $lead\text{-}time\text{-}w3$).

*Social sub-module*. Social sub-module simulates the people's psychological and behavioral response processes to the warnings. The $j$-th agent[1] will decide to

---

[1] The agent refers to the resident agent by default

evacuate when his/her overall evacuation intention ( $S_j$ , $S_j \in [0, 3]$ ) exceeds a
threshold, $\tau$ , or the water depth near him/her exceeds a threshold, $EDT$ . There are
two components in $S_j$ : evacuation intention arising from receiving warnings ( $S_j^W$ ,
$S_j^W \in \{1, 2, 3\}$ ), and evacuation intention arising from observing neighbors ( $S_j^N$ ,
$S_j^N \in [0, 1]$ ). The value of $S_j^W$ is related to the socio-demographic and socio-
psychological attributes of the $j$ -th agent ( $SSC_j$ ) and the stages of the receiving
warning from the early warning sub-module (*WT*). The relationship can be described
by a random forest algorithm. The value of $S_j^N$ equals to the proportion of the $j$ -th
agent's neighbors who have decided to evacuate. The weights of the influence of $S_j^W$
and $S_j^N$ on the $S_j$ are represented by parameters $\alpha_j$ and $\beta_j$ , respectively, and
$\alpha_j + \beta_j = 1$ . Finally, the overall evacuation intention of the $j$ -th agent at time $t$ , $S_{j,t}$ ,
is a linear combination of overall evacuation intention at time $t-1$ ( $S_{j,t-1}$ ) and current
information. Learning rate, $\theta_j$ , measures the weight given by the $j$ -th agent to the
obtained information at the current time. If the $j$ -th agent has decided to evacuate,
he/she will walk along the shortest road network to the shelters. His/her walking speed
is estimated by the spatial-grid evacuation model (SGEM) that has been developed by
the City University of Hong Kong and Wuhan University (Lo et al., 2004).
*Flood sub-module.* As flash flood can affect the people's evacuation behaviors and
cause casualties, the flash flood process is simulated in the flood sub-module. The
Hydrologic Engineering Center's River Analysis System (HEC-RAS) software is
gaining popularity due to its capabilities to simulate unsteady flow efficiently, and
identify and visualize flood-prone areas (Hicks and Peacock, 2005; Maidment, 2017).
The HEC-RAS model has been applied for flood forecasting and warning (Oleyiblo
and Li, 2010). And it has been adopted in our flood sub-module. The river geometries
such as centerlines, bank lines, and cross-sectional lines are the major parameters
proceeded in the HEC-RAS model to generate flood-prone areas. The spatiotemporal
changes in the depth and velocity of flash floods are simulated by the HEC-RAS model
after the warnings.
**2.1.3.    Casualty rate estimation module**
Current studies  generally  estimate  flood  casualties  through  two  types  of
influencing factors: environmental factors, and victim characteristics (Petrucci, 2022).
The first type includes the hazard conditions (measured by flood depth and velocity)
and the location and environments where the hazard occurs (e.g., urban/rural,
indoor/outdoor, and distance from floods). Flood velocity and depth are influenced by
underlying surface conditions, such as the topography of flood plains, watershed size,
and land use (Creutin et al., 2009; Penning-Rowsell et al., 2005; Spitalar et al., 2014).
Rural residents are more vulnerable to floods due to the lack of advanced emergency
response systems and forecasting and warning capabilities. The concentration of urban
population and the increase in impermeable surfaces will amplify the flood risk
(Brazdova and Riha, 2014; Terti et al., 2017). The second type includes the attributes
of people (e.g., age, gender, weight, and height), the status of the residence, and whether
the victim has taken adaptive or emergency measures (Papagiannaki et al., 2022;
Petrucci et al., 2019; Petrucci, 2022; Salvati et al., 2018).
Takahashi et al. (1992) established a connection between the characterization of
human stability (safe or fall) and flow features such as depth ($h$) and velocity ($u$)
through a casualty experiment. If variable $z$ is set to the linear addition of $h$ and $u$
(i.e., $z = \beta_0 + \beta_1 \times h + \beta_2 \times u$), a logistic regression equation can be used to fit the
relationship between the characterization of human stability (if the person falls, its value
is one, otherwise it is zero) and $z$. Based on the experiment data, the parameters ($\beta_0$,
$\beta_1$, and $\beta_2$) can be estimated, and the logistic regression equation will be used to
predict the probability of casualty by depth and velocity. Based on the spatiotemporal
distribution of the people outputted from the social sub-module and the spatiotemporal
distribution of floodwater outputted from the flood sub-module, the casualty probability
of an agent can be estimated via the logistic regression equation as follows:
$$f(z) = \frac{1}{1 + e^{15.48 - z}} \tag{1}$$

where $z = \beta_0 + \beta_1 \times h + \beta_2 \times u$, $\beta_0 = -12.37$, $\beta_1 = 22.036$, $\beta_2 = 11.517$. The flood
water depth is represented by $h$ ($h \in [0.28,\ 0.85]$ (m)), and the flood water velocity
is denoted by $u$ ($u \in [0.50,\ 2.00]$ (m/s)). The $j$-th agent is taken as casualty if the
$h$ exceeds 0.85 m or $u$ exceeds 2.00 m/s around him/her. The casualty rate is
estimated as the proportion of the casualties. A detail description of the ABM can be
retrieved from Zhang et al. (2024)
**2.1.4.   A surrogate model development for the ABM**
Due to the complexity of the ABM, running this model once requires a significant
amount of time (Confalonieri et al., 2010). To simulate multiple flash flood events, it is
necessary to improve the computational efficiency of the ABM. Thus, a Bayesian
method developed by Oakley and O'Hagan (2004) is used to develop a Gaussian
process (GP) emulation as a surrogate model of the ABM. The GP emulation can
simulate the warning response processes more efficiently than the original ABM
(O'Hagan, 2006). In general, the GP emulation can be represented by an equation:
$D = f_{GP}(\mathbf{x})$ where $D$ is the casualty rate at the end of the simulation and $\mathbf{x}$ are a set
of parameters of the ABM.
A global sensitivity analysis of the ABM reveals that the weight of warning
influence, $\alpha$, is the most sensitive parameter for the casualty rate (Zhang et al., 2024).
Furthermore, rainfall, $P$, is the driving factor causing flash floods. Therefore, if there
is a flash flood disaster and its corresponding warnings are issued, the ABM can be
simplified into a two-parameter surrogate model: $D = f_{GP}^2(\alpha, P)$. If there is a flash
flood disaster and no warning is issued, the ABM can be simplified into a one-parameter
surrogate model: $D = f_{GP}^1(P)$.
## 2.2.   Simulation chain of "rainstorm probability forecasting -
## decision on issuing warnings - warning response processes"
**2.2.1.   Simulation of the rainstorm probability forecasting**
Flash floods often occur if there are sufficient rainstorms in a small basin over a
few hours (Collier, 2007; Younis et al., 2008). As the total flood generation and routing
time is very short, flash flood warnings have to be dependent on the rainstorm
forecasting for an enough lead time (Zhai et al., 2018). Therefore, the rainstorm
forecasting determines the flash flood warning decisions. The probabilistic forecasting
is preferred over the deterministic one as it considers forecasting uncertainties and it is
beneficial for rational decisions (Krzysztofowicz, 2001). A random probabilistic
forecasting generator based on Ambühl (2010) is employed to forecast the probability
distribution of rainfall as follows:
$$F \sim N(P + N(\mu_{PA}, \sigma_{PA}^2), N(\mu_{PP}, \sigma_{PP}^2)) \tag{2}$$
where $F$ is the forecasted rainfall, $N(.)$ is the Gaussian distribution, $P$ is the
actual rainfall, $N(\mu_{PA}, \sigma_{PA}^2)$ reflects the forecasting accuracy, and $N(\mu_{PP}, \sigma_{PP}^2)$
reflects the forecasting precision.

Although Ambühl (2010) used the gamma distribution to simulate the forecasting

precision, the normal distribution can help improve the interpretability of the results. If
the probability distribution of forecasted rainfall is assumed to be normal distribution
and $\mu_{PA}$ is assumed to be zero according to Sawada et al. (2022), the deviation
between the median value of forecasted rainfall and the actual rainfall (denoted by $\eta$)
is determined by $\sigma_{PA}$. In other words, $\eta$ follows a normal distribution with a mean
of 0 and a variance of $\sigma_{PA}^2$. Therefore, there is a positive correlation between $|\eta|$ and
$\sigma_{PA}$. For example, assuming the actual rainfall is 0.5, if $\sigma_{PA} = 0.05$, the median value
of forecasted rainfall from each probability forecast is around 0.5. However, if
$\sigma_{PA} = 0.15$, the median value of forecasted rainfall is likely to deviate from 0.5 (see
**Figure 2a**). In fact, the probability of $\eta$ in the interval (-3$\sigma_{PA}$, 3$\sigma_{PA}$) is 99.73%.

Negative $N(\mu_{PP}, \sigma_{PP}^2)$ is truncated to $1.0 \times 10^{-6}$ to eliminate the negative values

of variance. The variance of forecasted rainfall is determined by $\mu_{PP}$. For example, the
probability distribution of forecasted rainfall is relatively concentrated if $\mu_{PP} = 0.1$
while the probability distribution of forecasted rainfall is relatively deconcentrated if
$\mu_{PP} = 0.2$ (see **Figure 2b**). And the variance of the variance of forecasted rainfall is
determined by $\sigma_{PP}$. As shown in **Figure 2c**, by conducting three probability forecasts,
there is a similar dispersion degree of probability distributions if $\sigma_{PP} = 0.01$ while
there is a distinguish dispersion degree of probability distributions if $\sigma_{PP} = 0.1$.

Briefly, if the mean of the $F$ (i.e., $P + N(0, \sigma_{PA}^2)$) is taken as the forecasting

tendency value, the accuracy of the forecasting tendency value will be reflected by $\sigma_{PA}$.
The variance of the $F$ (i.e., $N(\mu_{PP}, \sigma_{PP}^2)$) determines the band-width of the $F$. The
larger $N(\mu_{PP}, \sigma_{PP}^2)$, the greater the band-width value of the $F$. The variance of the
forecasting values is determined by $\mu_{PP}$, while the variance of the variance of the
forecasting values is determined by $\sigma_{PP}$.

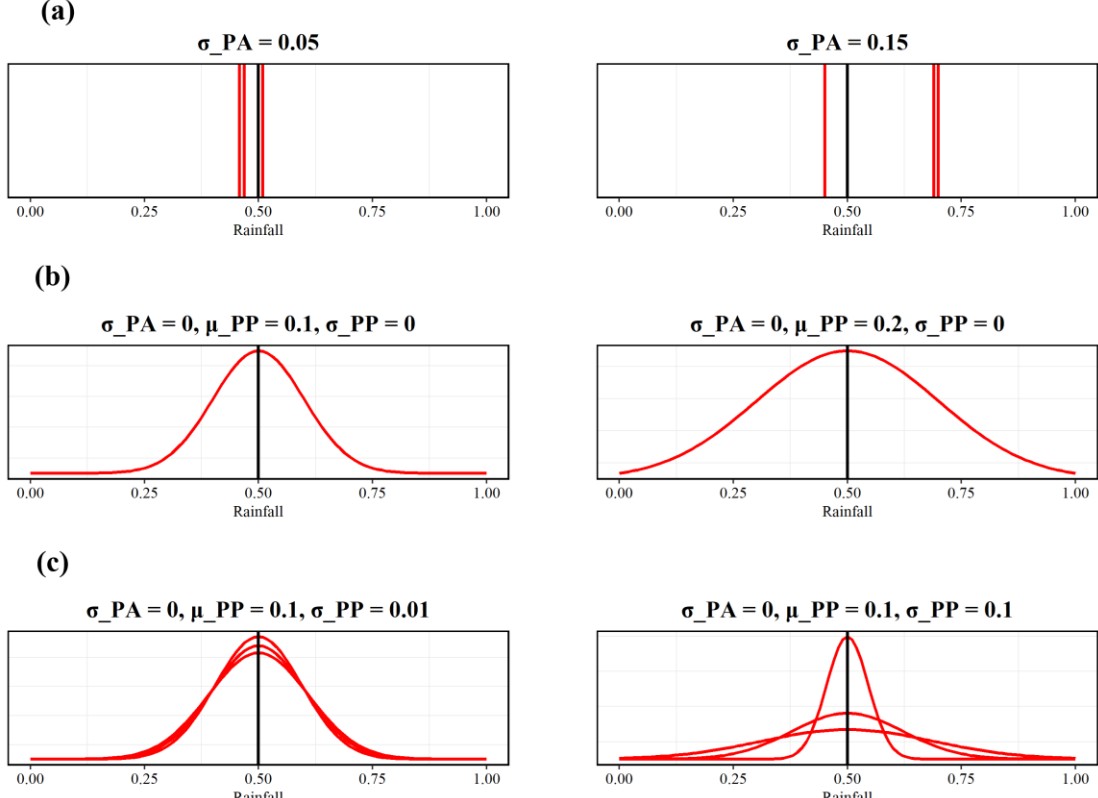

**Figure 2.** The black line represents the actual rainfall. The value of forecasted rainfall is normalized to 0-1. (a) The median value of forecasted rainfall (represented by the red lines) by conducting three probability forecasts under different $\sigma_{PA}$. (b) The probability distribution of forecasted rainfall (represented by the red line) under different $\mu_{PP}$. (c) The probability distributions of forecasted rainfall (represented by the red lines) by conducting three probability forecasts under different $\sigma_{PP}$.

### 2.2.2. Simulation of the decision on issuing warnings

There is a damage threshold, $\delta$. If the $P$ exceeds this threshold, flash flood disasters will occur and cause damages. The probabilistic forecasting system can provide the probability that the forecasted rainfall exceeds the $\delta$ (i.e., the probability of flash flood disasters, denoted by $Prob$). If the $Prob$ is larger than a preset threshold, $\lambda$, the warning administrators will issue the warnings. Thus, the $\lambda$ is the warning threshold. The warning outcomes are dependent on a contingency table (shown in **Table 1**). The outcomes are dependent on two conditions: first, whether the $Prob$ is above the $\lambda$ or not (i.e., whether to issue warnings or not); and second, whether the $P$ exceeds the $\delta$ or not (i.e., whether to occur a flash flood disaster or not). The interplay of the two conditions leads to four warning outcomes: true negative (no

warning), false negative (missed event), false positive (false warning), and true positive
(successful warning). The missed events and the false warnings are collectively taken
as failed warnings here.
**Table 1.** Contingency table defining the warning outcomes [a]

| | $P < \delta$ | $P \geq \delta$ |
|---|---|---|
| $Prob < \lambda$ | True negative (no warning) <br> *0* | False negative (missed event) <br> *Damage* |
| $Prob \geq \lambda$ | False positive (false warning) <br> *Cost* | True positive (successful warning) <br> *Cost + residual damage* |

[a] Costs and damages associated with each outcome. And they are highlighted in italics.
**2.2.3.    Simulation of the warning response processes**
According to the four warning outcomes in **Table 1**, the warning response
processes are simulated by the surrogate model of the ABM for estimating the casualty
rate, $D$. If the warning outcome is true negative or false positive, the casualty rate is
negligible as the actual rainfall, $P$, is smaller than the damage threshold, $\delta$. It should
be noted that false positive can cause opportunity cost as there are behavior responses
to the warnings (i.e., evacuation behaviors). As this study only focuses on the casualty
rate, the opportunity cost has been ignored. If the warning outcome is false negative,
there is a flash flood disaster but no warning is issued. In this case, the one-parameter
surrogate model (i.e., $D = f_{GP}^1(P)$) is employed to simulate the warning response
processes for estimating the casualty rate. If the warning outcome is true positive, there
is a flash flood disaster and its corresponding warnings are issued. The casualty rate is
mitigated by evacuation. The two-parameter surrogate model (i.e., $D = f_{GP}^2(\alpha, P)$) is
used to simulate the warning response processes for estimating the casualty rate. In
general, the casualty rate can be described by the following equation:
$$D = \begin{cases} 0 & \text{for true negative or false positve} \\ f_{GP}^1(P) & \text{for false negative} \\ f_{GP}^2(\alpha, P) & \text{for true positive} \end{cases} \quad (3)$$

We assume that past warning outcomes affect people's trust levels in the warnings.
Existing studies have found that the recent false warning ratio undermines people's trust
levels in the warnings and their preparedness actions (Jauernic and Van den Broeke,
2017; LeClerc and Joslyn, 2015; Lim et al., 2019; Ripberger et al., 2015). It is
reasonable to assume that people's past experiences with successful (or failed) warnings
increase (or decrease) their trust levels in the warnings. A person's trust level in the
warnings can be described by the parameter $\alpha$ representing the weight assigned to
the warning information. Therefore, $\alpha$ after experiencing a flash flood at the $t+1$
time can be described by the following equation:
$$\alpha(t+1) = \begin{cases} \alpha(t) & \text{for true negative} \\ \alpha(t) - \chi_{FN} & \text{for false negative} \\ \alpha(t) - \chi_{FP} & \text{for false positive} \\ \alpha(t) + \chi_{TP} & \text{for true positive} \end{cases} \tag{4}$$

where $\chi_{FN}$, $\chi_{FP}$, and $\chi_{TP}$ are increments of $\alpha$ for false negative, false positive,
and true positive, respectively. If $\alpha$ is larger than one, it is truncated to one. If $\alpha$ is
smaller than zero, it is truncated to zero. The people's trust levels in the warnings were
assumed to be only affected by the past warning outcomes. There are other factors (e.g.,
social education and government authority) that can be incorporate into the estimation
of the people's trust levels in further research.
**2.2.4.   Performance metrices of the warning**
Three metrics are used to evaluate the warning performance: the relative casualty
rate ($D_r$), missed event ratio ($MER$), and false warning ratio ($FWR$). The $D_r$ is
defined as:
$$D_r = \frac{D_w}{D_n} \tag{5}$$

where $D_w$ is the average casualty rate of multiple flash floods if there is a flash flood
warning. And the casualty rate of each flash flood can be estimated by equation (3).
$D_n$ is the average casualty rate of multiple flash floods if there is no flash flood
warning in place (i.e., the casualty rate is dependent only on the natural variability).
The casualty rate of each flash flood can be estimated by the following equation (6).
$$D_n = \begin{cases} 0 & \text{if } P < \delta \\ f_{GP}^1(P) & \text{if } P \geq \delta \end{cases} \tag{6}$$

The lower the value of $D_r$, the more effective the flash flood warning is. If the
objective of flash flood warning is the minimizing the casualties, the optimal warning
threshold is the threshold where the $D_r$ is the lowest.
Besides $D_r$, the $MER$ and $FWR$ are used to evaluate the performance of the
flash flood warning. They are defined by equations (7) and (8):
$$MER = \frac{O_{FN}}{O_{TP} + O_{FN}} \tag{7}$$

$$FWR = \frac{O_{FP}}{O_{FP} + O_{TP}} \qquad (8)$$

where $O_{FN}$, $O_{TP}$, $O_{FP}$ are the total number of false negative, true positive, and false
positive events, respectively.
## 3. Case study
### 3.1. Study area
Liulin Town located in Suixian Country, Hubei Province, China was selected as
our study area. The Lang River goes through Liulin Town as shown in **Figure 3(a)** and
the red rectangular box indicates the location of the town. The average annual rainfall
is 1,100 mm. Rainfall is unevenly distributed throughout the year, and mainly
concentrates from June to August. The upstream valley of Liulin Town is wider than
that of the downstream. And this river geomorphology hinders flood discharge and
easily causes the flash flood disaster when a rainfall occurs. Residences in the town are
located on both sides of Langhe River. In the prevention and control map of flash flood
disasters in Suixian County, two communities in Liulin Town are listed as high-risk and
relatively high-risk areas. Especially, an extreme rainfall with a volume of 503 mm
from 2:00 a.m. to 9:00 a.m. on August 12, 2021 (hereafter called the 8.12 event) caused
a severe flash flood disaster in the town. Unfortunately, 21 people were dead and four
people were still missing in this disaster although flash flood warnings had been issued
(Wei, 2021). Exploring the way to determine the threshold of issuing flash flood
warnings in the town will provide valuable information on flash flood disaster
prevention for reducing the casualties.

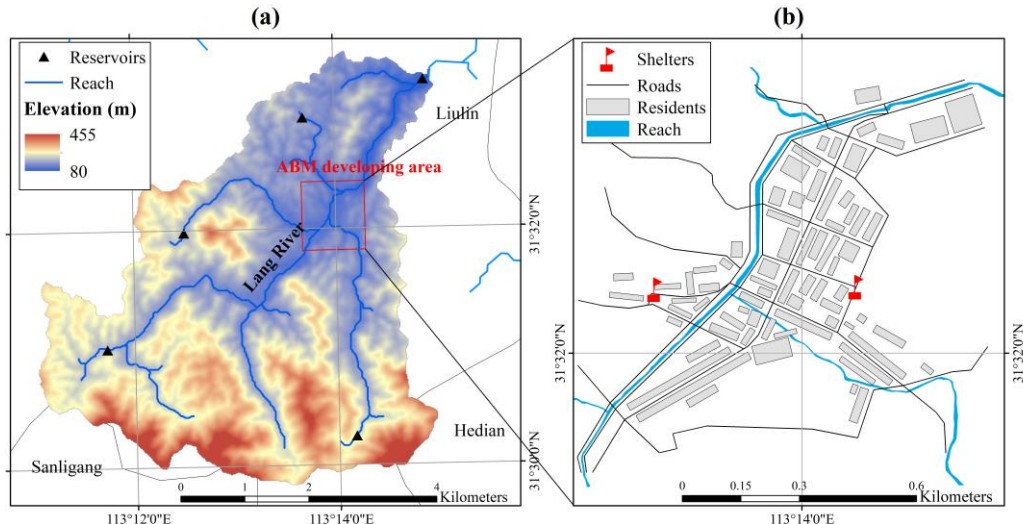


**Figure 3.** Location of the (a) Lang River Basin and (b) Liulin Town

## 3.2.    Setting of the ABM

To set up the environment of the ABM, the residences and road network (see **Figure 3**) were imported into the model after processing a digital archive (i.e., World Imagery Wayback). To prevent evacuation across the river, two shelters were set up at high place on both sides of the Langhe River. And they should not be submerged by floods. The parameters of the ABM were set according to calibration, empirical data, and related literature (see **Table 2**). The lead times of the three stages of warning and evacuation depth threshold were parameterized from the two-month surveying expertise and experience in the study area. The lead time of rainstorm red warning is around 180 min in China, and here the lead time was set to 120 min as a conservative and unfavorable scenario. As people should immediately move to a shelter after receiving an immediate-evacuation warning, the lead time of immediate-evacuation warning is related to the travel time of the people to the shelter. The person farthest from the shelter needs about 25 min to travel to the shelter, so the lead time of immediate-evacuation warning was set to 30 min. According to the lead times of rainstorm red warning and immediate-evacuation warning, it was assumed that the lead time of ready-to-evacuate warning was between the two, that is, 60 min. The three hyperparameters of the random forest model were calibrated by the empirical data from our survey. A sampling without replacement was conducted on the empirical data and the sample was used to assign the initial $SSC$ values of the agents. The random forest model calibration, the survey, and the method of assigning $SSC$ values were detailed in Zhang et al. (2024). The values of $\theta_j$ and $p_j$ of the $j$-th agent were sampled from the Gaussian distributions according to the exiting literature (Du et al., 2017). The setting of these two parameters aimed to reflect people's general behavior. $\beta_j = 0.5$ represents a general and unbiased behavior that gives same weights to current flood information and past opinion on flood risk. And $p_j = 0.1$ means flood information being checked every ten minutes. $S_j = 2$ is set to indicate no decision making on evacuation for the $j$-th agent in the empirical data while $S_j > 2$ means the evacuation decision of the agent. Hence, the value of $\tau$ was set to 2. A global sensitivity analysis has been performed to explore the relative impacts of these parameters on the casualty rate and can be retrieved from Zhang et al. (2024).

**Table 2.** Fixed ABM parameters

| Sub-module | Parameters | Symbol | Values | Remark |
|---|---|---|---|---|
| Early warning | Lead time of rainstorm red warning | *lead-time-w*1 | 120 min | Author estimation [a] |
| | Lead time of ready-to-evacuate warning | *lead-time-w*2 | 60 min | Author estimation [a] |
| | Lead time of immediate-evacuation warning | *lead-time-w*3 | 30 min | Author estimation [a] |
| Random forest | Number of trees | *ntree* | 500 | Calibration |
| | Number of candidate variables | *mtry* | 6/1/6 [b] | Calibration |
| | Minimum size of nodes | *nodesize* | 10/1/10 [b] | Calibration |
| | Socio-demographic and socio-psychological characteristics of resident agents | *SSC* | | Empirical data |
| Opinion dynamics | Learning rate | $\theta$ | 0.5 (0.1) [c] | Literature reference (Du et al., 2017) |
| | Probability of receiving early warnings | *p* | 0.1 (0.1) [c] | Literature reference (Du et al., 2017) |
| | Evacuation threshold | $\tau$ | 2 | Empirical data |
| Others | Visual range | *VR* | 40 m | Literature reference (Wu et al., 2022) |
| | Evacuation depth threshold | *EDT* | 0.28 m | Author estimation [a] |

[a] These estimations are from the two-month surveying expertise and experience of the authors
in the study area. [b] $x_1/x_2/x_3$ indicates the values of the factors are $x_1$, $x_2$, and $x_3$ for the rainstorm
red, the ready-to-evacuate, and the immediate-evacuation warnings, respectively. [c] $x_1$ ($x_2$)
indicates the values of the factors are sampled from a normal distribution with mean value of
$x_1$ and variance of $x_2$
The flood-module of the ABM was formed by a two-dimensional (2D)
hydrodynamic model in the Langhe River Basin through HEC-RAS. Terrain
information was obtained from the digital elevation model (DEM) at a spatial resolution
of 12.5 m provided by the Advanced Land Observing Satellite (ALOS). Cells with size
of 30 m were generated within the 2D flow areas. The Manning's coefficient was set to
a unified comprehensive value of 0.045. The upstream boundary condition was set as
the rainfall process. The hyetograph was selected by the measured rainfall process of
the 8.12 event. Specifically, the hourly rainfall was greater than 30.0 mm from 2:00 to
7:00 on August 11, 2021 and the 6-h rainfall was up to 462.6 mm (see **Figure 4**). The
6-h rainfall process was input into the HEC-RAS as the hyetograph. As Baiguo River

reservoir is in the outlet, the downstream boundary condition was set as the normal
water level of the reservoir. The spatiotemporal changes in the depth and velocity of
flash floods were exported after running the model at a temporal interval of 2 min and
spatial resolution of 12.5 m. it should be noted that the hyetograph was selected as the
measured rainfall process of the 8.12 event. More uneven hyetographs should be taken
in the flash flood simulation, and the impact of hyetograph on the warning threshold
determination can be explored in further research.

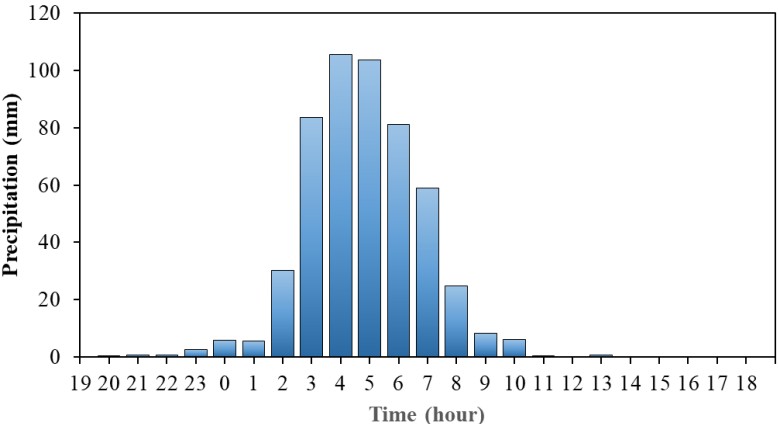

**Figure 4.** The rainfall process from 19:00 on August 11 to 19:00 on August 12, 2021 of
Liulin Meteorological Station

The ABM was run by covering the processes from issuing warnings to flash flood
at a time step of 1 min and spatial resolution of 9.6 m. And 500 agents were assumed
to be involved in the simulations. Due to the inherent randomness of the ABM, the
averages of the outputs from the repeating 1,000 times for running the ABM were
obtained to ensure stable outputs.

## 3.3. Rainfall data

A series of rainfall data was imported into the ABM for simulating a series of
possible flash flood disasters. First, synthetic rainfall series were generated to ensure
the representative of the extreme events. The annual maximum 6-h rainfall, $P$, was
assumed to follow the Pearson III distribution. Its values of mean and $C_v$ in the basin
above Liulin Town were estimated to be 80 mm and 0.6, respectively, according to Atlas
of Statistical Parameters of rainfall in Hubei Province (2008). $C_s / C_v$ was taken as 3.5
in Hubei Province. A total of 1,000 synthetic rainfall events were randomly generated
by the Pearson III distribution, and the result was shown in **Figure 5**. Second, a rainfall
event in the synthetic rainfall events was input into the flood module of ABM, and then

converted into a flash flood event. According to the flash flood event, the degree of
flash flood disaster had been estimated, and people's attitudes towards the
corresponding warning had been recorded. The people's attitudes can influence the
subsequent warning response processes. Then, the next rainfall event in the synthetic
rainfall events was input into the ABM, and the above simulation process was repeated.

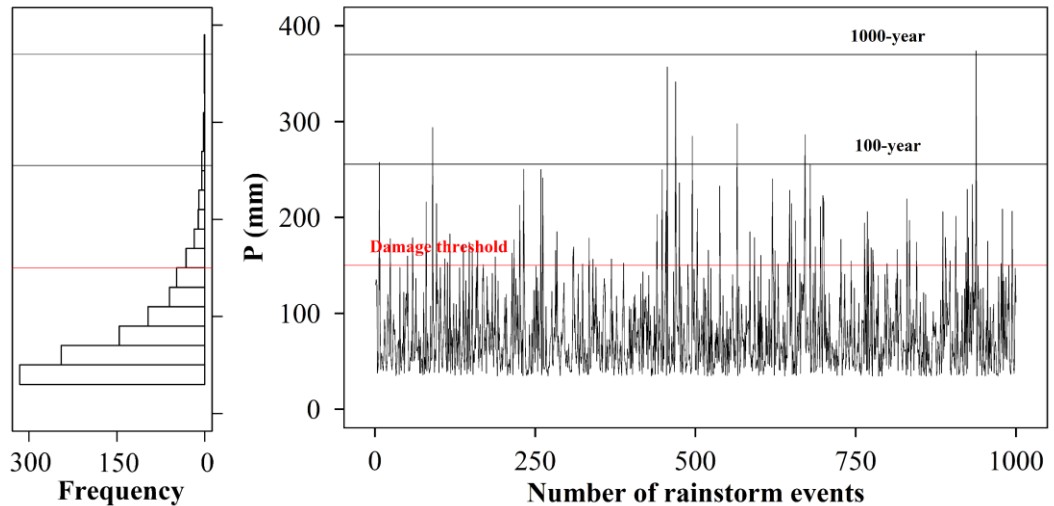


**Figure 5.** 1,000 synthetic series of rainfall events (right). Histogram statistical results
of the synthetic rainfall events. The three horizontal lines from top to bottom represent
the rainfall for 1000-year return period, 100-year return period, and triggering disasters,
respectively

## 3.4. Model test experiments

The impact of forecasting skills on the warning threshold determination can be
explored by setting different values of $\sigma_{PA}$, $\mu_{PP}$, and $\sigma_{PP}$. In real-world flood
warning scenarios, these three parameters can be estimated by statistical methods, such
as moment estimation method and maximum likelihood estimation method.
Specifically, the actual rainfall and the corresponding probability forecasting results in
the history can be collected under a certain forecasting skill. Each rainstorm event is
taken as a sample, and the observed rainfall, the median value of probability forecasted
rainfall, and the variance of probability distribution for the rainstorm event are
estimated. By collecting multiple rainstorm events, these three parameters can be
estimated using statistical methods for a certain forecasting skill. As we aim to examine
the uncertainties in flash flood forecasting that affect the determination of warning
thresholds in this study, three possible values of each of the three parameters (i.e., $\sigma_{PA}$,
$\mu_{PP}$, and $\sigma_{PP}$) were prepared to reflect different forecasting skills (see **Table 3**) and
their interactive effects on the determination of warning threshold were tested.

Rainstorm red warning is the highest level of meteorological risk warning in the

mainland of China. When the rainstorm red warning is issued, floods tend to cause
damage and the residents in flood risk area are advised to evacuate (Wang et al., 2020).
If the 6-hour rainfall is up to 150 mm, the rainstorm red warning will be issued
(Shanghai Meteorological Bureau, 2019). Thus, the value of $\delta$ was taken as 150 mm
in the case study.
**Table 3.** Model test experiment for determining the warning threshold under different
forecasting skills

| Parameters | Symbol | Values |
|---|---|---|
| The accuracy of the forecasting tendency value | $\sigma_{PA}$ | {0.05, 0.10, 0.15} |
| The variance of the forecasting values | $\mu_{PP}$ | {0.0, 0.1, 0.2} |
| The variance of the variance of the forecasting values | $\sigma_{PP}$ | {0.0, 0.1, 0.2} |
| Damage threshold | $\delta$ | 150 mm |
| Increment of $\alpha$ for false negative | $\chi_{FN}$ | 0.1 |
| Increment of $\alpha$ for false positive | $\chi_{FP}$ | 0.1 |
| Increment of $\alpha$ for true positive | $\chi_{TP}$ | 0.1 |

Besides the uncertainties of the forecasting, there are uncertainties in people's

response processes to the uncertain forecasting. To determine the warning threshold
under different forecasting skills and tolerance levels of the failed warnings, the
warning threshold was determined under different $\sigma_{PA}$ and combinations of
parameters related to the increments of $\alpha$ (i.e., $\chi_{FN}$, $\chi_{FP}$, and $\chi_{TP}$) through Exp1
in **Table 4**, and under different $\mu_{PP}$ and combinations of parameters related to the
increments of $\alpha$ through Exp 2 in **Table 4**. The higher the $\chi_{FN}$ and $\chi_{FP}$, the lower
the tolerance levels of the people towards the missed event and the false warnings,
respectively.
**Table 4.** Model test experiment for determining the warning threshold under different
forecasting skills and tolerance levels of the failed warnings

| Parameters | Symbol | Values | |
|---|---|---|---|
| | | Exp1 | Exp2 |
| The accuracy of the forecasting tendency value | $\sigma_{PA}$ | {0.05, 0.10, 0.15} | 0.075 |
| The variance of the forecasting values | $\mu_{PP}$ | 0.15 | {0.0, 0.1, 0.2} |
| The variance of the variance of the forecasting values | $\sigma_{PP}$ | 0.075 | 0.075 |
| Damage threshold | $\delta$ | 150 mm | 150 mm |

| Parameters | Symbol | Values | |
|---|---|---|---|
| | | Exp1 | Exp2 |
| Increments of $\alpha$ for false negative, false positive, and true positive | $\chi_{FN}/\chi_{FP}/\chi_{TP}$ | {0.1/0.1/0.1, 0.8/0.8/0.1, 0.8/0.1/0.1, 0.1/0.8/0.1} | {0.1/0.1/0.1, 0.8/0.8/0.1, 0.8/0.1/0.1, 0.1/0.8/0.1} |

## 4. Results and discussions

### 4.1. The casualty rate from people's response process simulation

To determine the warning threshold based on the people's response process simulation, the ABM with different values of $P$ and $\alpha$ were run to generate corresponding casualty rates, and these simulations were taken as sample data to train the GP emulation as a surrogate model of the ABM, as shown in **Figure 6**. And it has shown the variation of casualty rate with $\alpha$ under different $P$. There are three stages of change in the casualty rate as $\alpha$ increases regardless of $P$. When $\alpha$ increases from 0.0 to 0.4, the casualty rate slowly decreases; but as $\alpha$ continues to increase to 0.6, the rate of decline becomes faster. When $\alpha$ is greater than or equal to 0.6, everyone arrives at the shelters before the flash flood disaster arrives and there are no casualties regardless of $P$. This result implies that it is very important and effective to enhance people's trust levels in the warnings when people have similar trust levels in warning information and their neighbors. When people's trust in warning information decreases, their evacuation decisions will become more dependent on whether their neighbors are evacuating or not. In other words, the increase in the overall evacuation intention ($S$) of agents requires their neighbors to take evacuation actions. However, taking evacuation actions requires the increase in $S$ in turn. Thus, waiting for others' evacuation ultimately leads to neither an increase in $S$ nor the implementation of evacuation actions.

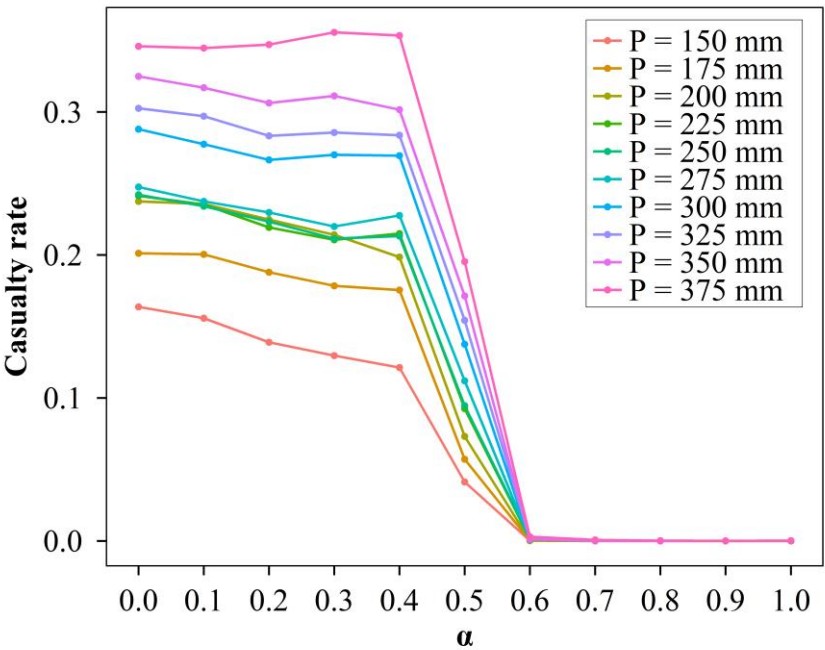

**Figure 6.** The casualty rate under different values of $P$ and $\alpha$ from ABM simulations

Because the casualty rate is zero when $\alpha$ is greater than or equal to 0.6 regardless of $P$, the one-parameter and two-parameter GP emulations were trained for $\alpha$ with a value less than 0.6 and the results were shown in **Figure 7**. The training result for one-parameter GP emulation shows that there are also three stages in the increase of casualty rate as $P$ increases. When $P$ increases from 150 to 200 mm, the casualty rate increases; but if $P$ increases from 200 to 260 mm, the casualty rate remains almost unchanged. When $P$ exceeds 260 mm and continues to increase, the casualty rate starts to increase again. This result indicates that there is spatial heterogeneity of flood risk levels in the case study. It is necessary to classify flood risk zones and distinguish water level or rainfall thresholds for triggering evacuation according to different flood risk levels. The training result for two-parameter GP emulation shows the complex responses of casualty rate to changes in $\alpha$ and $P$. When $\alpha$ is less than 0.4, there are three stages of changes in the casualty rate as $P$ increases. As $\alpha$ increases from 0.4 to 0.6, the relationship between $P$ and casualty rate tends to be linearly positive, and the difference in casualty rates under different $P$ gradually reduces. This result means that the trust level in the warnings becomes the dominant factor in determining the casualty rate when the people's trust levels in the warnings and their neighbors are similar (i.e., when the value of $\alpha$ is the range of 0.4 to 0.6).

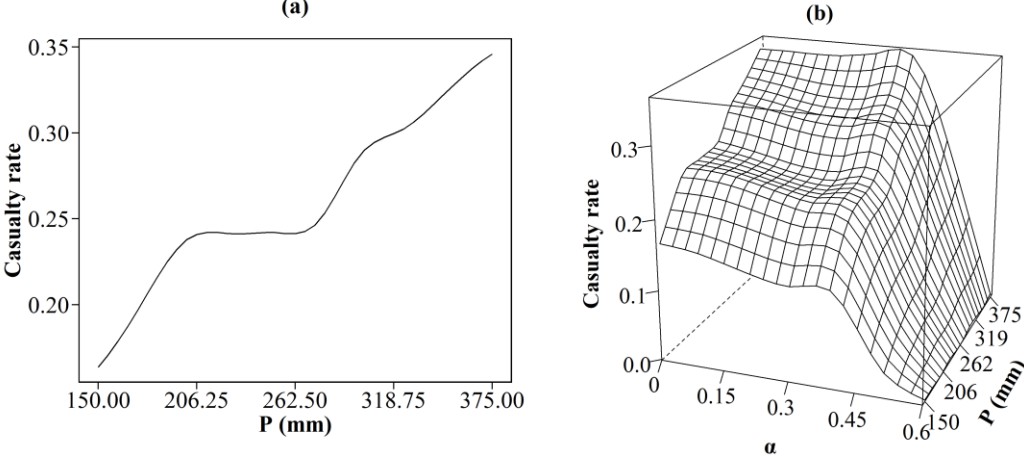


**Figure 7.** Trained (a) one-parameter and (b) two-parameter GP emulations for casualty
rate

## 4.2. Determining the warning threshold under different forecasting skills for minimizing casualties

To determine the warning threshold under different forecasting skills for
minimizing casualties, 250-member Monte Carlo simulations were performed on the
simulation chain of "rainstorm probability forecasting - decision on issuing warnings -
warning response processes" by randomly perturbing the warning threshold, $\lambda$, under
different values of parameters controlling the forecasting skills (see **Figure 8**). Different
rows represent different values of $\mu_{PP}$, and there is a larger forecasting variance in the
sub-graph of the lower row. Similarly, there is a larger variance of the forecasting
variance in the sub-graph of the right column compared to the sub-graph of the left
column. The highest forecasting accuracy is represented by the green curves, followed
by the yellow curves, and finally the red curves. In all the sub-graphs, there is the
highest relative casualty rate in the red curves, followed by the yellow curves, and
finally the green curves. Therefore, the lower the forecasting accuracy, the higher the
relative casualty rate. The optimal warning threshold can be taken as the value of $\lambda$
where the relative casualty rate, $D_r$ is lowest. The optimal warning thresholds are the
lowest in the green curves, followed by the yellow curves, and finally the red curves in
all the sub-graphs. Thus, the lower the forecasting accuracy, the higher the optimal
warning threshold. The reasons can be found in **Figure 9**. As the warning threshold
decreases, the number of false warnings and successful warnings increases, and more
warnings are issued. However, if the forecasting accuracy is low, the proportion of false
warnings is higher than that of successful warnings among the additional warnings
issued. For example, as the warning threshold decreases, the green curve for low
forecasting accuracy rises faster than that for high forecasting accuracy. This means that
if the forecasting accuracy is low, as the warning threshold decreases, the increase speed
of false warnings is higher than that of successful warnings. In addition, when the
warning threshold is less than 0.7, the green curve begins to rise rapidly for $\sigma_{PA} = 0.15$,
while it does not start to rise rapidly until the warning threshold is less than 0.5 for
$\sigma_{PA} = 0.15$. Therefore, when the forecasting accuracy is low, a high warning threshold
should be set. As the forecasting accuracy increases, lowering the warning threshold
can result in more successful warnings without significantly increasing false warnings,
thereby improving the effectiveness of flash flood warnings.

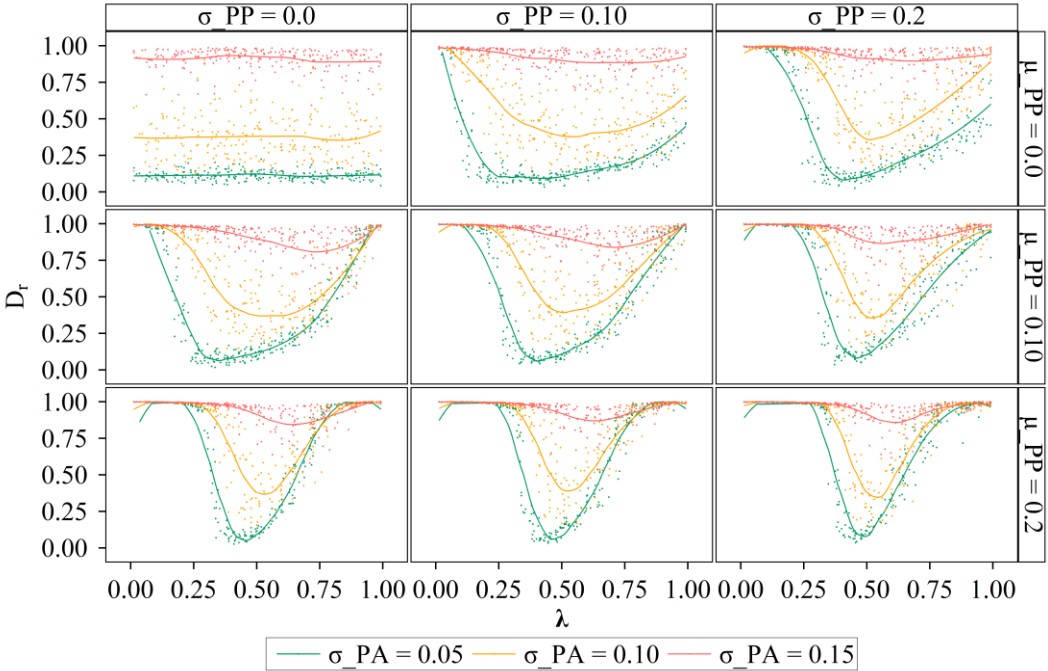


**Figure 8.** The relationship between the relative casual rate, $D_r$, and the warning
threshold, $\lambda$, under different values of $\sigma_{PA}$, $\mu_{PP}$, and $\sigma_{PP}$. Different rows and
columns represent different values of $\mu_{PP}$ and $\sigma_{PP}$, respectively. Different colors
represent different values of $\sigma_{PA}$. Each dot shows the result of the individual Monte
Carlo simulation

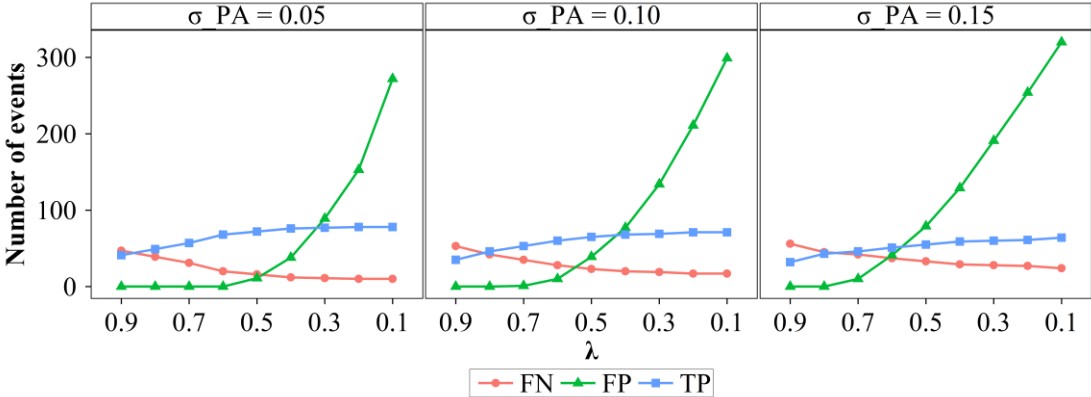

**Figure 9.** The changes in the number of false negative, false positive, and true positive events as warning threshold decreases, $\lambda$ under different values of $\sigma_{PA}$. The range of $\lambda$ is reversed from 0.9 to 0.1

In terms of the impacts of the forecasting variance (see **Figure 8**), there is a larger forecasting variance and a higher relative casualty rate of three colored curves in the sub-graph of the lower row. Thus, the larger the forecasting variance, the higher the relative casualty rate. For the optimal warning threshold, the differences in the optimal warning thresholds of these three colored curves are smaller in the sub-graph of the lower row. For instance, as the forecasting variance increases, the optimal warning thresholds for the red curves decrease while the optimal warning thresholds for the green curves increase. This result means that the larger the forecasting variance, the lower the optimal warning threshold for low forecasting accuracy, while the larger the forecasting variance, the higher the optimal warning threshold for high forecasting accuracy. When the forecasting accuracy is at a low level, a large forecasting variance is actually beneficial for improving the forecasting skills. High forecasting skill means that more successful warnings and fewer false warnings are issued after lowering the warning threshold. Therefore, if the forecasting accuracy is at a low level, as the forecasting variance increases, the warning threshold can be lowered. On the contrary, if the forecasting accuracy is at a high level, as the forecast variance increases, increasing the warning threshold can significantly decrease the false warnings and improve the effectiveness of flash flood warnings. Finally, we focused on the impacts of the variance of the forecasting variance. Similar to the impacts of the forecasting variance, the larger the variance of the forecasting variance, the higher the relative casualty rate. As the variance of the forecasting variance increases, the optimal warning threshold tends to decrease for low forecasting accuracy or to increase for high

forecasting accuracy.

The impacts of the three parameters (i.e., $\sigma_{PA}$, $\mu_{PP}$, and $\sigma_{PP}$) on the shape of

the relationship curve between $D_r$ and $\lambda$ can be analyzed as follows. As shown in
**Figure 8**, $\sigma_{PA}$ determines the height of the curve, while $\mu_{PP}$ and $\sigma_{PP}$ determine
the width of the curve. Specifically, as the forecasting accuracy increases, the stationary
point of the curve moves down and the curve becomes higher; as the forecasting
variance or the variance of the forecasting variance increases, the curve becomes
narrower. If the forecasting accuracy is high and the forecasting variance and the
variance of the forecasting variance are large, the curve will become high and narrow,
such as the green curve for $\mu_{PP} = 0.2$ and $\sigma_{PP} = 0.2$. And there is only a low relative
casualty rate near the optimal warning threshold in this green curve. Thus, it is more
important to determine the optimal warning threshold for minimizing casualties if the
forecasting accuracy is higher, and the forecasting variance and the variance of the
forecasting variance are larger.
**4.3.  Determining the warning threshold under different forecasting**
**skills and tolerance levels of the failed warnings for minimizing**
**casualties**

To determine the warning threshold under different forecasting skills and tolerance

levels of the failed warnings for minimizing casualties, the simulation chain of
"rainstorm probability forecasting - decision on issuing warnings - warning response
processes" was run with random values of $\lambda$ under different $\sigma_{PA}$ and combinations
of parameters related to the increments of $\alpha$ (i.e., $\chi_{FN}$, $\chi_{FP}$, and $\chi_{TP}$) (see **Figure**
**10**), and different $\mu_{PP}$ and combinations of parameters related to the increments of
$\alpha$ (i.e., $\chi_{FN}$, $\chi_{FP}$, and $\chi_{TP}$) (see **Figure 11**). Owing to the similar roles of $\mu_{PP}$, and
$\sigma_{PP}$, the effects of $\sigma_{PP}$ on the determination of warning threshold were not explored
here. As shown in **Figure 10**, the optimal warning thresholds for the yellow curves are
the lowest. The yellow curves represent scenarios that people's trust in warnings is
sensitive to false negative events and people have a low tolerance level for the missed
events. To reduce the missed event ratio, the warning threshold should be lowered (see
**Figure 10g**). Therefore, the warning threshold should be lowered for increasing
people's trust levels in warnings and reducing casualties if people have a lower
tolerance level for the missed events. Similarly, the warning threshold should be
increased if the people's tolerance levels for the false warnings become lower (see the
red curves). And if the people's tolerance for both the missed events and the false
warnings decreases to the same level, the optimal warning threshold remains almost
unchanged, but the relative casualty rate overall increases (see the blue curves). As for
the relative casualty rate, the relative casualty rates of the yellow curves are lower than
those of the red curves. This result suggests that compared to the missed events, the
people's low tolerance levels for the false warnings are less conducive to the
effectiveness of flash flood warnings. As shown in **Figure 9**, the number of false
warnings is greater than the number of missed events in general. Therefore, if the
people's tolerance levels for the false warnings is low, their trust levels in warnings are
more likely to decrease, leading to the effects of "cry wolf".

By comparing **Figure 10a** and **Figure 10b**, the overall height of the curves

decreases when the forecasting accuracy decreases, as discussed in the last paragraph
of section 4.2. However, compared to green curve, the heights of other curves decrease
more significantly. And the relative casualty rates are high at any warning threshold
(i.e., $D_r > 0.75$) except for the green curve when the $\sigma_{PA}$ increases from 0.05 to 0.1.
It is more pronounced when the $\sigma_{PA}$ further increases to 0.15. Therefore, as the
forecasting accuracy decreases, the benefits gained by adjusting the warning threshold
based on the people's tolerance levels of the failed warnings decreases. In other words,
no matter how the warning threshold is adjusted, the relative casualty rate is high and
the effectiveness of warning is at a low level.

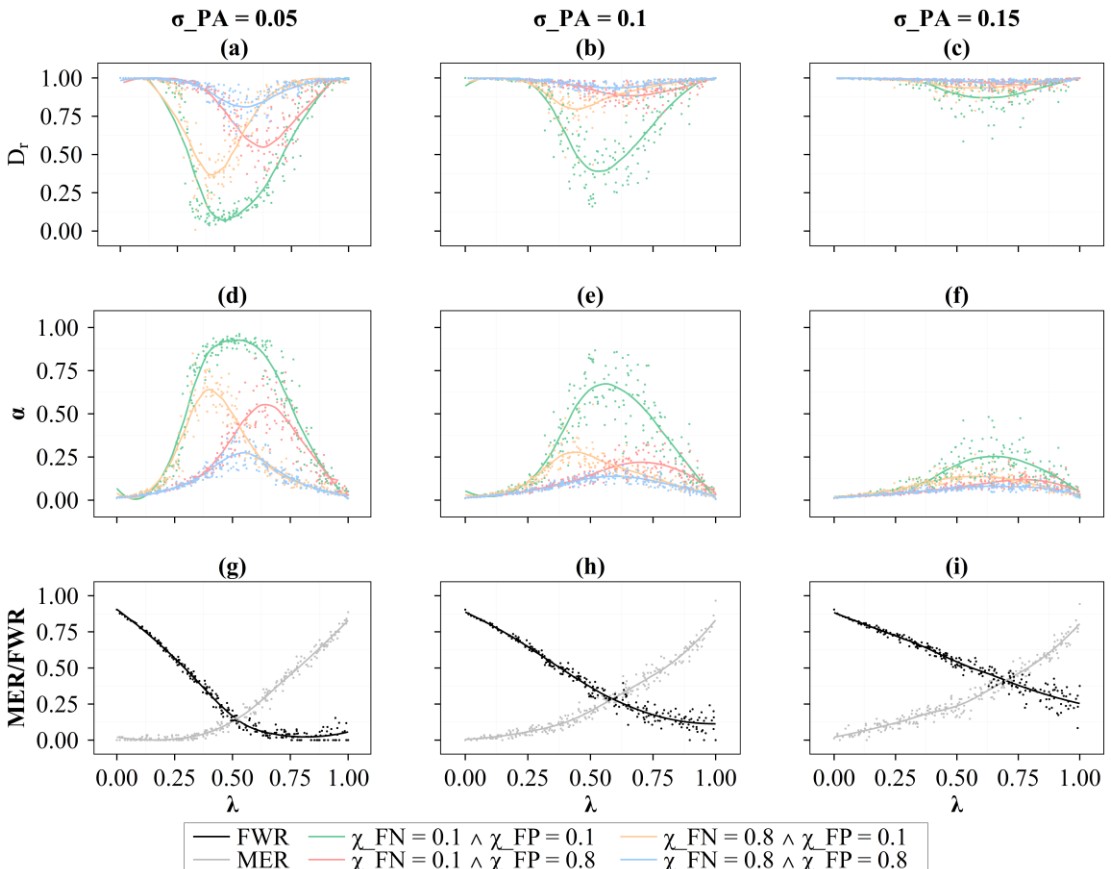

**Figure 10.** (a-c) The relationship between the warning threshold, $\lambda$ and the relative casualty rate, $D_r$ under different $\sigma_{PA}$ and combinations of parameters related to the increments of $\alpha$ (i.e., $\chi_{FN}$, $\chi_{FP}$, and $\chi_{TP}$). (d-f) Same as (a-c) but for time-averaged $\alpha$. (g-i) The relationship between the warning threshold, $\lambda$, and the false warning ratio, $FWR$, and the missed event ratio, $MER$, under different $\sigma_{PA}$. Each dot shows the result of the individual Monte Carlo simulation

In terms of the effects of the forecasting variance and the tolerance levels of the failed warnings on the determination of warning threshold as shown in **Figure 11**, the warning threshold should be decreased if people have a lower tolerance level for the missed events, and vice versa. And compared to the missed events, the people's low tolerance levels for the false warnings are less conducive to the effectiveness of flash flood warnings. These findings are consistent with the results in **Figure 10**. Furthermore, we find that the difference in the optimal warning thresholds of these colored curves decreases as the forecasting variance increases as shown in **Figure 11a-Figure 11c**. As discussed in the last paragraph of section 4.2, the curve becomes narrower as the forecasting variance increases. If the width of the curves decreases, the difference between their optimal warning thresholds will also decrease. Therefore, as the

forecasting variance increases, the difference in the optimal warning thresholds of these
curves will decrease, and the adjustment space for the warning threshold based on the
people's tolerance levels will also decrease.

If the green curve represents the result of the baseline scenario where both $\chi_{FN}$

and $\chi_{FP}$ equal 0.1, increment of the values of $\chi_{FN}$ and $\chi_{FP}$ (i.e., lowering
tolerance levels for the missed events and the false warnings) will result in a series of
curves, and these curves will be enveloped by the green curve in **Figure 11**. Therefore,
only when the green curve is high enough, can the relative casualty rate of this series
of curves be low enough, and the effectiveness of flash flood warnings be sufficiently
improved. And only when the green curve is wide enough, can the difference in the
optimal warning threshold for this series of curves be large enough, and there can be
enough room for adjustment the warning threshold. In summary, by increasing the
height and width of the green curve, the adjustable room for the warning threshold will
be larged and the effectiveness of flash flood warnings will be improved. As the
forecasting accuracy increases, the green curve becomes higher. And as the forecasting
variance decreases, the green curve becomes wider. Therefore, under the premise of
improving the forecasting skills (i.e., increasing forecasting accuracy and decreasing
forecasting variance), adjusting the warning threshold based on the people's tolerance
levels of the failed warnings is one of the ways to improve the effectiveness of flash
flood warnings.

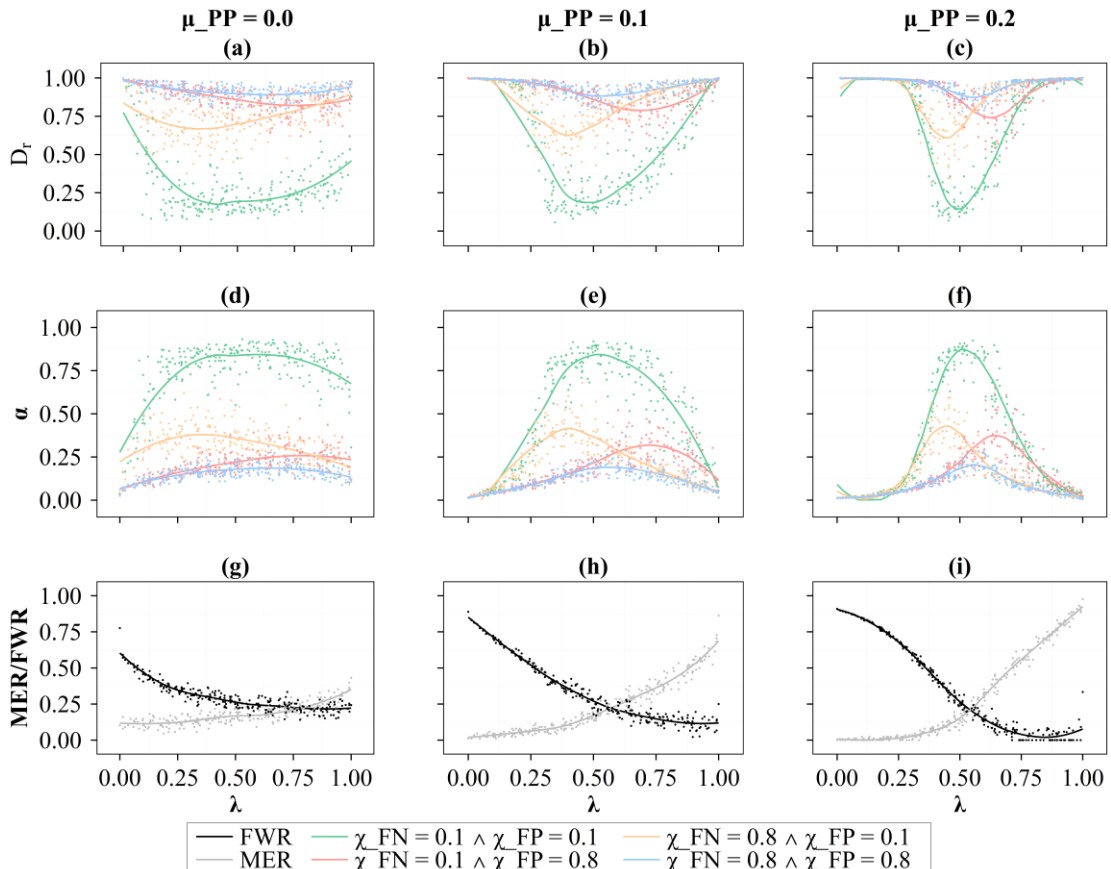

**Figure 11.** (a-c) The relationship between the warning threshold, $\lambda$ and the relative casualty rate, $D_r$ under different $\mu_{PP}$ and combinations of parameters related to the increments of $\alpha$ (i.e., $\chi_{FN}$, $\chi_{FP}$, and $\chi_{TP}$). (d-f) Same as (a-c) but for time-averaged $\alpha$. (g-i) The relationship between the warning threshold, $\lambda$, and the false warning ratio, $FWR$, and the missed event ratio, $MER$, under different $\mu_{PP}$. Each dot shows the result of the individual Monte Carlo simulation

## 4.4. Implication and limitations

Although the simulation results have deepened our understanding of the warning threshold determination, especially the impact of forecasting skills and people's tolerance levels of the failed warnings on the warning threshold determination, the simulation results should be carefully interpreted due to the assumptions underlying the simulation method. As highlighted in the simulation results, the warning threshold should be appropriately determined due to the trade-off between multiple factors affecting the warning threshold (see **Figure 12**). Specifically, as the warning threshold increases, the number of missed events and the loss of $\alpha$ due to missed events will increase. And as the missed events increase, the level of disaster preparedness will decrease. The loss of $\alpha$ and the low level of disaster preparedness are not conductive

to reducing disaster damage. However, as the warning threshold increases, the number
of false warnings and the loss of $\alpha$ due to false warnings will decrease, which is
conductive to reducing disaster damage. Therefore, there is a trade-off in the warning
threshold determination. However, it has been assumed that the experience of warnings
(i.e., the success or failure of past warnings) only affects people's trust levels in
warnings (i.e., $\alpha$). Actually, the experience of warnings can also affect people's
attitudes and behaviors towards flash floods. Specifically, the dangerous experiences
on the property/life losses can form deep flash flood memories. The damage memories
make people more inclined to evacuate after receiving warnings (Cuite et al., 2017;
Morss et al., 2018). The higher the warning threshold, the more missed events and
dangerous experiences there will be, and people's damage memories will be more
profound. The profound damage memories increase people's evacuation intention and
reducing disaster damage. Therefore, if combined with the dynamism of human
behaviors, there still be a trade-off of the warning threshold determination but the
optimal warning threshold will increase.

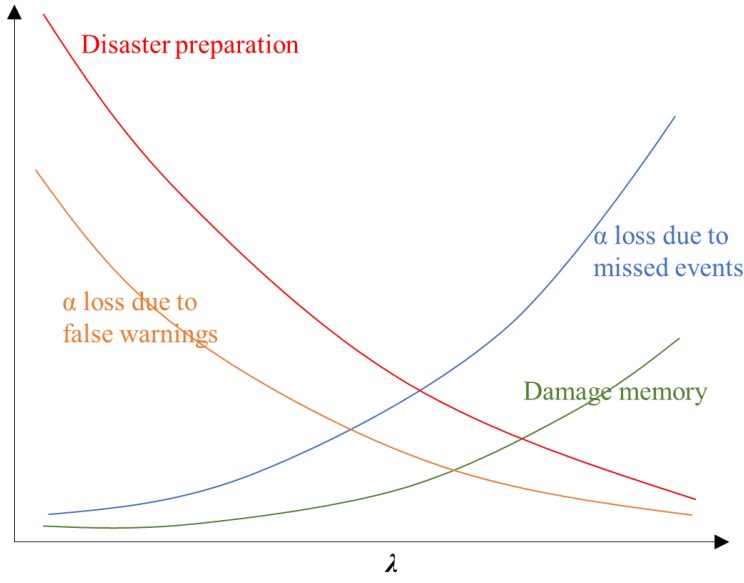


**Figure 12.** A schematic diagram that illustrates the trade-off in the warning threshold
determination
The development of the ABM is the core of the simulation flow. The simulation
results based on the ABM show that there is a monotonic positive relationship between
$\alpha$ and casualty rate (see **Figure 7**). The rationale behind the monotonic relationship is
that the higher the value of $\alpha$, the more likely a person is to evacuate after receiving a
warning. If someone has evacuated, he/she will lead more people to evacuate, because
neighbor behavior is an important information source for a person to make evacuation
decisions. The developed ABM generalizes these two information sources (i.e., warning
information and neighbor behavior) to simulate the processes of people's evacuation
decision making. However, environmental cue (e.g., rainfall condition) is also an
information source (Lindell et al., 2019). The monotonic positive correlation
relationship between $\alpha$ and casualty rate may no longer hold true if the environmental
cue is incorporated in the ABM. For example, if there is a flash flood disaster but no
warning is issued, our ABM assumes that no one will evacuate. In fact, if people
observe the rainfall that may lead to flash flood disasters, they will evacuate even if no
warning is issued. The high trust levels in warnings ($\alpha$) may have suppressed their
evacuation intention, leading to a higher casualty rate instead. If the monotonic positive
correlation relationship between $\alpha$ and casualty rate no longer holds true, the curve
shape in **Figure 8** will no longer be unimodal, and the determination of the optimal
warning threshold will become more complex.

The ABM was applied to Liulin Town where residences are located along Lang
River and listed as high-risk and relatively high-risk areas. If there is a flash flood
disaster, the whole town along the river is likely to be submerged and all the people are
required to evacuate. Therefore, the modeling region with an area of 0.28 km$^2$ is set as
a whole to receive forecasting and warnings. However, if study region is large and
terrain is complex, the study region needs to be divided into multiple sub-regions and
then modeled by the ABM accordingly. For each sub-region, forecasting and warnings
also need to be produced and issued separately. However, in real world, there is usually
a lack of clarity of the sub-region impact of some of the warnings owing to the limitation
of forecasting skills. Forecasting and warning often only target a certain region and are
difficult to distinguish the different degrees of impact within that region (Roberts et al.,
2022). Given a unified forecast and warning for a region, the sub-region along river or
at high-risk areas is prone to missed events, while the sub-region located on a high
ground is prone to false warnings. If it is difficult to improve forecasting skills,
modifying people's tolerance levels of the failed warnings will become one of the ways
to improve the effectiveness of warnings. For example, education or risk
communication can be conducted to inform residents of the background and production
process of warning information, allowing them to understand the reasons for false
warnings and missed events, as well as the obstacles to eliminate these issues.
Implementing targeted education or risk communication based on geographical location
to adjust people's tolerance for corresponding types of failed warnings can compensate
for the lack of accuracy in forecasting and warning.
It is a tough work to verify the hydrodynamic simulation and people's evacuation
process simulation in small watersheds due to the difficulty in collecting data. The field
flood survey was used to verify the water depth simulated by HEC-RAS. The flood
survey showed that the flood depth of high-rise houses was 1.75 m, while that of houses
with low terrain was 3.85 m in the 8.12 event (Shaojun et al., 2022). The survey results
are roughly consistent with our simulation. In further studies, technologies such as
unmanned aerial vehicle and radar can be used to obtain high-precision inundation data,
and the simulation results can be finely verified based on the inundation data. For the
verification of the evacuation processes simulated by the social sub-module in the ABM,
indirect verification was conducted by investigating and simulating people's evacuation
intention. To directly verifying the evacuation process simulation, milling time can be
surveyed and then converted into data on the evacuation processes in further studies.
Based on the data, the parameters of the social sub-module in the ABM can be
calibrated and verified.
**5. Conclusions**
A method has been proposed to determine the warning threshold for minimizing
casualties based on the people's response process simulation. A process-based ABM
was developed to simulate people's response processes to flash flood warnings. A
simulation chain of "rainstorm probability forecasting - decision on issuing warnings -
warning response processes" was conducted to determine the warning threshold based
on the ABM. The main conclusions are as follows.
The casualty rate is jointly controlled by the warning information source and
precipitation. If the people's trust levels in official warnings are below a certain
threshold, precipitation is the dominant factor in controlling the casualty rate. If the
people have a similar level of trust in official warnings and neighbor behaviors, the
credibility of the warning information source is the dominant factor in controlling the
casualty rate.
The warning threshold has been determined under different forecasting skills for
minimizing casualties. The lower the forecasting accuracy, the higher the optimal
warning threshold. And the larger the forecasting variance or the variance of the

forecasting variance, the higher (lower) the optimal warning threshold for high (low) forecasting accuracy. Furthermore, the impact pattern of forecasting skills on the shape of the relationship curve between the relative casualty rate and the warning threshold has been revealed: the curve becomes higher as the forecasting accuracy increases, and the curve becomes narrower as the forecasting variance or the variance of the forecasting variance increases.

The warning threshold has been determined under different forecasting skills and tolerance levels of the failed warnings for minimizing casualties. The warning threshold should be decreased (increased) if people have a lower tolerance level for the missed events (the false warnings). However, if the forecasting accuracy is low and the forecasting variance is large, the space for adjusting the warning threshold is limited, and no matter how the warning threshold is adjusted, the casualty rate remains at a high level, and the effectiveness of flash flood warnings is limited. Therefore, under the premise of improving the forecasting skills, adjusting the warning threshold based on the people's tolerance levels of the failed warnings is one of the ways to improve the effectiveness of flash flood warnings.

## Code availability

The code that supports the findings of this study is available from the corresponding author upon reasonable request.

## Date availability

Data will be made available on request.

## Author contribution

Ruikang Zhang: Conceptualization, Formal analysis, Methodology, Writing – original draft, Visualization, Funding acquisition. Dedi Liu: Conceptualization, Data curation, Formal analysis, Funding acquisition, Methodology, Supervision, Writing - review & editing. Lihua Xiong: Project administration, Supervision. Jie Chen: Data support, Methodology, Writing - review & editing. Hua Chen: Validation, Writing - review & editing, Supervision. Jiabo Yin: Validation, Writing - review & editing. All authors contributed to the interpretation of the results and to the text.

## Competing interests

The authors declare that they have no conflict of interest.

## Disclaimer

Publisher's note: Copernicus Publications remains neutral with regard to jurisdictional claims in published maps and institutional affiliations.

## Acknowledgments

The authors gratefully acknowledge the financial support from National Key Research and Development Project of China (2022YFC3202803), the National Natural Science Foundation of China (52379022), and the Open Innovation Foundation funded by ChangJiang Survey, Planning, Design and Research Co., Ltd (CX2021K04).

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
