# Peer review of "Determining the threshold of issuing flash flood"

_Hydrology and Earth System Sciences, 2024_

## Author Comment (AC1)

**Reviewer #1**

People's response to flood warnings is an important factor that affect the performance of flood evacuation processes. This study develops an agent-based model to simulate the people's response processes to the warnings, and to determine the threshold for issuing flood warnings. The Liulin Town in China is selected to analyze the role of flood warning threshold and forecast variance in flood fatality rates. The modeling results provide interesting insights into effective flood management. Overall, this is a well conducted research with clear presentations. Below are some minor comments:

Response: Thank you very much for your comments that have significantly improved the quality of this study. In the following, we have detailed how these comments (in black) are raised and our responses (in dark blue).

**Comment 1**

Table 3 lists three parameters to represent flood forecast skills. Please add some text to describe the meaning of these parameters, and how to quantify these parameters in real-world flood warning scenarios.

**Response 1**

Thank you for the valuable comment. Indeed, these three parameters are highly generalized, and there is insufficient explanation to help readers intuitively understand their meanings and how to quantify their values.

If the probability distribution of forecasted rainfall is assumed to be normal distribution, the deviation between the median value of forecasted rainfall and the actual rainfall (denoted by $\eta$) is determined by $\sigma_{PA}$. In other words, $\eta$ follows a normal distribution with a mean of 0 and a variance of $\sigma_{PA}^2$. Therefore, there is a positive correlation between $|\eta|$ and $\sigma_{PA}$. For example, assuming the actual rainfall is 0.5, if $\sigma_{PA} = 0.05$, the median value of forecasted rainfall from each probability forecast is around 0.5. However, if $\sigma_{PA} = 0.15$, the median value of forecasted rainfall is likely to deviate from 0.5 (see **Figure 1**). In fact, the probability of $\eta$ in the interval ($-3\sigma_{PA}$, $3\sigma_{PA}$) is 99.73%.

The variance of forecasted rainfall is determined by $\mu_{PP}$. For example, the probability distribution of forecasted rainfall is relatively concentrated if $\mu_{PP} = 0.1$ while the probability distribution of forecasted rainfall is relatively deconcentrated if $\mu_{PP} = 0.2$ (see **Figure 2**). And the variance of the variance of forecasted rainfall is determined by $\sigma_{PP}$. As shown in **Figure 3**, by conducting three probability forecasts, there is a similar dispersion degree of probability distributions if $\sigma_{PP} = 0.01$ while there is a distinguish dispersion degree of probability distributions if $\sigma_{PP} = 0.1$.

In real-world flood warning scenarios, these three parameters can be estimated by statistical methods, such as moment estimation method and maximum likelihood estimation method. Specifically, the actual rainfall and the corresponding probability forecasting results in the history can be collected under a certain forecasting skill. Each rainstorm event is taken as a sample, and the observed rainfall, the median value of probability forecasted rainfall, and the variance of probability distribution for the rainstorm event are estimated. By collecting multiple rainstorm events, these three parameters can be estimated using statistical methods for a certain forecasting skill.

We will add the above discussion to the manuscript.

[Figure]

**Figure 1.** The median value of forecasted rainfall (represented by the red lines) by conducting three probability forecasts under different $\sigma_{PA}$. The black line represents the actual rainfall. The value of forecasted rainfall is normalized to 0-1

[Figure]

[Figure]

**Figure 2.** The probability distribution of forecasted rainfall (represented by the red line) under different $\mu_{PP}$. The black line represents the actual rainfall. The value of forecasted rainfall is normalized to 0-1

[Figure]

[Figure]

**Figure 3.** The probability distributions of forecasted rainfall (represented by the red lines) by conducting three probability forecasts under different $\sigma_{PP}$. The black line represents the actual rainfall. The value of forecasted rainfall is normalized to 0-1

**Comment 2**

The model is quite complex with a lot of parameters. A modeling framework diagram is needed to show all the model components, the associated parameters and their relationships.

**Response 2**

Thank you for the constructive comment. We will add a modeling framework that determines the warning threshold based on people's response processes to the manuscript. The modeling framework includes the development of an ABM and its surrogate model for simulating the people's response processes to flash flood warnings and a chain simulation of "forecasting – warning - response" (see **Figure 4**). First, rainstorm probability forecasting is performed according to actual rainfall. And then the warning administrators make decisions to issue warnings based on the rainstorm probability forecasting and warning thresholds. If it is decided to issue warnings, the warning information and the actual rainstorm jointly drive the surrogate model of ABM to simulate the people's response processes. Finally, the casualty rate is estimated and

the warning threshold that minimizes the casualty rate can be determined based on the proposed modeling framework.

[Figure]

**Figure 4.** The proposed modeling framework for determining the warning threshold based on people's response processes (the parameters in a simulation step are indicated by a rectangular box with the corresponding color background)

**Comment 3**

Equation (1) describes the fatality probability as a function of flood water depth and flood water velocity. Where does this equation come from? A concise literature review on flood causality function could be helpful to make the paper more solid.

**Response 3**

Thanks for your comments. The equation comes from the experimental results of Takahashi et al. (1992). Current studies generally estimate flood casualties through two types of influencing factors: environmental factors, and victim characteristics (Petrucci, 2022). The first type includes the hazard conditions (measured by flood depth and

velocity) and the location and environments where the hazard occurs (e.g., urban/rural, indoor/outdoor, and distance from floods) (Creutin et al., 2009; Penning-Rowsell et al., 2005; Spitalar et al., 2014). The second type includes the attributes of people (e.g., age, gender, weight, and height), the status of the residence, and whether the victim has taken adaptive or emergency measures (Papagiannaki et al., 2022; Petrucci et al., 2019; Petrucci, 2022; Salvati et al., 2018). We will provide a detailed review to discuss the estimation of flood casualties.

**References:**

Creutin, J. D., Borga, M., Lutoff, C., Scolobig, A., Ruin, I., and Créton-Cazanave, L.: Catchment dynamics and social response during flash floods: the potential of radar rainfall monitoring for warning procedures, Meteorol. Appl., 16, 115-125, https//doi.org/10.1002/met.128, 2009.

Papagiannaki, K., Petrucci, O., Diakakis, M., Kotroni, V., Aceto, L., Bianchi, C., Brázdil, R., Gelabert, M. G., Inbar, M., Kahraman, A., Kiliç, Ö., Krahn, A., Kreibich, H., Llasat, M. C., Llasat-Botija, M., Macdonald, N., de Brito, M. M., Mercuri, M., Pereira, S., Rehor, J., Geli, J. R., Salvati, P., Vinet, F., and Zêzere, J. L.: Developing a large-scale dataset of flood fatalities for territories in the Euro-Mediterranean region, FFEM-DB, Sci. Data, 9, https//doi.org/10.1038/s41597-022-01273-x, 2022.

Penning-Rowsell, E., Floyd, P., Ramsbottom, D., and Surendran, S.: Estimating injury and loss of life in floods: A deterministic framework, Nat. Hazards, 36, 43-64, https//doi.org/10.1007/s11069-004-4538-7, 2005.

Petrucci, O.: Review article: Factors leading to the occurrence of flood fatalities: a systematic review of research papers published between 2010 and 2020, Nat. Hazards Earth Syst. Sci., 22, 71-83, https//doi.org/10.5194/nhess-22-71-2022, 2022.

Petrucci, O., Aceto, L., Bianchi, C., Bigot, V., Brázdil, R., Pereira, S., Kahraman, A., Kiliç, Ö., Kotroni, V., Llasat, M. C., Llasat-Botija, M., Papagiannaki, K., Pasqua, A. A., Rehor, J., Geli, J. R., Salvati, P., Vinet, F., and Zêzere, J. L.: Flood Fatalities in Europe, 1980-2018: Variability, Features, and Lessons to Learn, Water, 11, https//doi.org/10.3390/w11081682, 2019.

Salvati, P., Petrucci, O., Rossi, M., Bianchi, C., Pasqua, A. A., and Guzzetti, F.: Gender, age and circumstances analysis of flood and landslide fatalities in Italy, Sci. Total Environ., 610, 867-879, https//doi.org/10.1016/j.scitotenv.2017.08.064, 2018.

Spitalar, M., Gourley, J. J., Lutoff, C., Kirstetter, P. E., Brilly, M., and Carr, N.: Analysis of flash flood parameters and human impacts in the US from 2006 to 2012, J. Hydrol., 519, 863-870, https//doi.org/10.1016/j.jhydrol.2014.07.004, 2014.

Takahashi, S., Endoh, K., and Muro, Z. I.: Experimental study on people's safety against overtopping waves on breakwaters, rep. 31-04 The Port and Harbour Res, Inst., Yokosuka, Japan, 20, 1992.

---

## Author Comment (AC2)

**Reviewer #2**

The manuscript "Determining the threshold of issuing flash flood warnings based on people's response process simulation" by Zhang et al. presents a novel and comprehensive approach to determining flash flood warning thresholds that considers the complexities of human response processes, which is a significant advancement over traditional methods that often focus solely on natural hydrological processes. Additionally, the study examines the uncertainties in flash flood forecasting that affect the effectiveness of warning thresholds and the role of people's tolerance for false warnings and missed events in setting these thresholds. The manuscript is generally well-written, and I have only a few suggestions for the authors to consider before it can be accepted for publication.

**Response:** Thank you very much for this positive assessment. We have followed your comments and revised the manuscript carefully. Please see the point-by-point responses to your comments as follows.

**Comment 1**

While the introduction highlights the limitations of existing approaches, it lacks explicit statements of the specific research questions the study aims to address. I would suggest that the authors improve the introduction to more clearly frame the study's objectives and guide the reader through the subsequent sections.

**Response 1**

Thanks for the insightful comments. We have revised the last paragraph of the introduction to clarify the research objectives, especially the relationship between research objectives and the limitations of existing approaches. The modified content is as follows:

*The objective of this study includes two parts. Firstly, to simulate people's response processes to flash flood warnings and reveal the impact of the warning information weight given by people on the effectiveness of warnings, this study aims to develop a process-based ABM that combines natural and social processes (section 2.1). Secondly, to determine the threshold of issuing warnings (called*

*warning threshold hereafter) based on the social processes of warning responses, this study attempts to propose a simulation chain of "rainstorm probability forecasting - decision on issuing warnings - warning response processes" based on the ABM (section 2.2). Through the proposed simulation framework for determining the warning threshold, we will examine the uncertainties in flash flood forecasting that affect the determination of warning thresholds and the joint impact of forecasting skills and people's tolerance levels of failed warnings on the warning threshold determination. Liulin Town in China is selected as a case study to demonstrate the proposed method, and to provide valuable insights into the determination of warning threshold for improving the effectiveness of flash flood warnings.*

**Comment 2**

For the methodology section, I strongly recommend adding a diagram or flowchart to illustrate the relationships between the different modules and the overall simulation chain, including key variables and processes. This will be helpful for the readers to quickly understand the complex interactions and flow of information within the model.

**Response 2**

Thank you for the constructive comment. We have added a modeling flowchart that presents the simulation chain of "forecasting – warning - response" (see **Figure R1**) in the revised manuscript for illustrating the relationship between key processes/variables. A detailed description of the figure has also been given in the revised manuscript as follows:

*First, rainstorm probability forecasting is performed according to actual rainfall. And then the warning administrators make decisions to issue warnings based on the rainstorm probability forecasting and warning thresholds. If it is decided to issue warnings, the warning information and the actual rainstorm jointly drive the surrogate model of ABM to simulate the people's response processes. Finally, the casualty rate is estimated and the warning threshold that minimizes the casualty rate can be determined based on the proposed modeling framework.*

[Figure]

**Figure R1.** The proposed modeling framework for determining the warning threshold based on people's response processes (the parameters in a simulation step are indicated by a rectangular box with the corresponding color background)

**Comment 3**

In the methods, some parameters are estimated based on author expertise and empirical data. I would suggest more information to justify the parameter settings. For example, although I assume that the parameters in section 2.1.3 are validated in the authors' previous study, at least the rationale behind the choice should be clearly articulated. In addition, it is not clear why the casualty probability is estimated using a logistic regression equation based on flood water depth and velocity. Please clarify the rationale for the choice of this particular equation.

**Response 3**

Thank you for the insightful comment.

For the casualty rate estimation module, we have added an explanation regarding the rationale for the choice of the logistic regression equation and the setting of its parameters in the revised manuscript. The detailed explanation in the revised manuscript is as follows:

*Takahashi et al. (1992) established a connection between the characterization of human stability (safe or fall) and flow features such as depth (h) and velocity (u) through a casualty experiment, and the result is shown in **Figure R2**. If variable z is set to the linear addition of h and u (i.e., $z = \beta_0 + \beta_1 h + \beta_2 u$), a logistic regression equation can be used to fit the relationship between the characterization of human stability (if the person falls, its value is one, otherwise it is zero) and z (see **Figure R3**). Based on the experiment data, the parameters ($\beta_0$, $\beta_1$, and $\beta_2$) can be estimated, and the logistic regression equation will be used to predict the probability of casualty by depth and velocity.*

[Figure]

**Figure R2.** Casualty experiment - results of Takahashi et al. (1992) experiment

**Figure R3.** Casualty estimation - binomial logistic regression derived from the experimental data of Takahashi et al. (1992)

The lead times of the three stages of warning were parameterized based on author expertise, and parameters *ß* and *p* were set based on existing literature. Therefore, we have also added the rationale behind the parameter setting in the revised manuscript as follows:

*The lead times of the three stages of warning were parameterized from the two-month surveying expertise and experience in the study area. Specifically, the lead time of rainstorm red warning is around 180 min in China, and here the lead time was set to 120 min as a conservative and adverse scenario. As people should immediately move to a shelter after receiving an immediate-evacuation warning, the lead time of immediate-evacuation warning is related to the travel time of the people to the shelter. The person farthest from the shelter needs about 25 min to travel to the shelter, so the lead time of immediate-evacuation warning was set to 30 min. According to the lead times of rainstorm red warning and immediate-evacuation warning, it was assumed that the lead time of ready-to-evacuate warning was between the two, that is, 60 min. The three hyperparameters of the random forest model were calibrated by the empirical data from our survey.*

*The setting of parameters β and p were based on existing literature, aiming to reflect people's general behavior. β = 0.5 represents a general and unbiased behavior that gives same weights to current flood information and past opinion on flood risk. And p = 0.1 means flood information being checked every ten minutes.*

**Comment 4**

In the case study section, it is not clear what the role of event rainfall and synthetic rainfall events are in ABM. Also, if possible, please try to improve Figure 3 to make it more informative, e.g., use a histogram and indicate the level of real event rainfall in it.

**Response 4**

Thanks for this helpful comment. The rainfall is the driving data of the flood module of ABM. After synthetic rainfall series was generated through statistical methods, the rainfall events in the synthetic rainfall series were input into the flood module of ABM in sequence, and previous rainfall events will have an impact on subsequent rainfall events. A detailed explanation that has been added in the revised manuscript is as follows:

*A rainfall event in the synthetic rainfall events was input into the flood module of ABM, and then converted into a flash flood event. According to the flash flood event, the degree of flash flood disaster had been estimated, and people's attitudes towards the corresponding warning had been recorded. The people's attitudes can influence the subsequent warning response processes. Then, the next rainfall event in the synthetic rainfall events was input into the ABM, and the above simulation process was repeated.*

The **Figure 3** in the original manuscript has been modified to include rainfall for different return periods and rainfall threshold causing disasters, as shown in **Figure R4**. The return period of rainfall in the 8.12 event (i.e., 462.6 mm) exceeds 1,000 years. And the frequency of the synthetic rainfall events was summarized through histogram plot. There is a decreasing trend of the frequency with increasing rainfall. The number of rainfall event reaching the damage threshold and exceeding the return period of 100 years and 1,000 years were only 88, nine, and one, respectively.

[Figure]

**Figure R4.** 1,000 synthetic series of rainfall events (right). Histogram statistical results of the synthetic rainfall events. The three horizontal lines from top to bottom represent the rainfall for 1000-year return period, 100-year return period, and triggering disasters, respectively

**Comment 5**

The results focus heavily on the simulation results without sufficient discussion of the implications and limitations of the results. For example, while the simulation results are detailed, there appears to be limited validation of these results against real-world data or historical flood events. Also, the results rely heavily on behavioral assumptions embedded in the ABM, such as confidence levels and evacuation intentions; I would suggest discussing the limitations of these assumptions and how they might affect the simulation results. In addition, the manuscript assumes that most parameters in the ABM are time-invariant, with the exception α. This simplification may overlook the dynamic nature of human behavior and environmental conditions. It would be beneficial to explore how varying these parameters over time might affect the model results.

**Response 5**

Thanks for the valuable comments. We acknowledge that the discussion on the limitations and implications of the research results in the paper is weak. Thus, we have added a new section 4.4 called "implication and limitations" in the revised manuscript. In the section, the impact of time-varying parameters on the trade-off in the warning threshold determination will be discussed. Afterwards, the limitations of the

assumptions in the model structure (i.e., generalization of disaster information sources) and their impact on the warning threshold determination will be explained. Finally, some considerations for model verification and future verification prospects will be presented. The detailed content of the section that has been supplemented is as follows:

[revised manuscript text omitted]

**Comment 6**

Although the case study describes the town's geomorphology and flood risk areas, it does not clearly link these local conditions to the model results and findings. Please consider to discuss how specific local conditions (e.g., topography, infrastructure) influenced the simulation results and how these findings can be generalized to other regions with similar or different conditions.

**Response 6**

Thanks for the constructive comments. We have added a paragraph in section 4.4 to discuss how to integrate specific local conditions into ABM simulation and the impact of different local conditions on our research findings. The detailed discussion in the revised manuscript is as follows:

*The ABM was applied to Liulin Town where residences are located along Lang River and listed as high-risk and relatively high-risk areas. If there is a flash flood disaster, the whole town along the river is likely to be submerged and all the people are required to evacuate. Therefore, the modeling region with an area of*

*0.28 km$^2$ is set as a whole to receive forecasting and warnings. However, if study region is large and terrain is complex, the study region needs to be divided into multiple sub-regions and then modeled by the ABM accordingly. For each sub-region, forecasting and warnings also need to be produced and issued separately. However, in real world, there is usually a lack of clarity of the sub-region impact of some of the warnings owing to the limitation of forecasting skills. Forecasting and warning often only target a certain region and are difficult to distinguish the different degrees of impact within that region (Roberts et al., 2022). Given a unified forecast and warning for a region, the sub-region along river or at high-risk areas is prone to missed events, while the sub-region located on a high ground is prone to false warnings. If it is difficult to improve forecasting skills, modifying people's tolerance levels of the failed warnings will become one of the ways to improve the effectiveness of warnings. For example, education or risk communication can be conducted to inform residents of the background and production process of warning information, allowing them to understand the reasons for false warnings and missed events, as well as the obstacles to eliminate these issues. Implementing targeted education or risk communication based on geographical location to adjust people's tolerance for corresponding types of failed warnings can compensate for the lack of accuracy in forecasting and warning. We will add the above discussion in the revised manuscript.*